# Effects of Stroke Rehabilitation Using Gait Robot-Assisted Training and Person-Centered Goal Setting: A Single Blinded Pilot Study

**DOI:** 10.3390/healthcare11040588

**Published:** 2023-02-16

**Authors:** Yeongmi Ha, Mingyeong Park

**Affiliations:** 1College of Nursing, Institute of Health Sciences, Gyeongsang National University, Jinju 52727, Republic of Korea; 2Yeson Rehabilitation Medicine Hospital, Jinju 52717, Republic of Korea

**Keywords:** stroke, rehabilitation, gait robot, goal setting

## Abstract

Many stroke survivors have difficulties due to the mobility and activities required in daily living. A walking impairment negatively affects the independent lifestyle of stroke patients, requiring intensive post-stroke rehabilitation. Therefore, the purpose of this study was to examine the effects of stroke rehabilitation using gait robot-assisted training and person-centered goal setting on mobility, the activities of daily living, stroke self-efficacy, and health-related QoL in stroke patients with hemiplegia. An assessor-blinded quasi-experimental study with a pre-posttest nonequivalent control group was used. Participants who were admitted to the hospital with a gait robot-assisted training system were assigned to the experimental group, and those without gait robots were assigned to the control group. Sixty stroke patients with hemiplegia from two hospitals specialized in post-stroke rehabilitation participated. Stroke rehabilitation using gait robot-assisted training and person-centered goal setting for stroke patients with hemiplegia was conducted for a total of six weeks. There were significant differences between the experimental group and control group in the Functional Ambulation Category (t = 2.89, *p* = 0.005), balance (t = 3.73, *p* < 0.001), Timed Up and Go (t = −2.27, *p* = 0.027), Korean Modified Barthel Index (t = 2.58, *p* = 0.012), 10 m Walking test (t = −2.27, *p* = 0.040), stroke self-efficacy (t = 2.23, *p* = 0.030), and health-related quality of life (t = 4.90, *p* < 0.001). A gait robot-assisted rehabilitation using goal setting for stroke patients with hemiplegia improved gait ability, balance ability, stroke self-efficacy, and health-related quality of life in stroke patients.

## 1. Introduction

Following the rapid aging of the Korean population, the prevalence of stroke in the population over 65 years old has increased from 4.6% in 1998 to 6.6% in 2020 [1]. Despite declining mortality rates from stable stroke incidence, the prevalence of stroke survivors with disabilities has been increasing [2]. According to a population-based study of disability after stroke in the UK, 40% of stroke survivors are disabled between one month and five years after stroke [3]. In Republic of Korea, approximately 9–10% of stroke survivors almost completely recover, and 65–70% of them have shown minor to severe impairments, including speech, swallowing, vision, ambulation, cognition, and coordination [4]. Stroke patients show decreased walking, balance, and daily activity performance caused by lower limb weakness, difficulty controlling movement, and spasticity [5]. Such a problem negatively affects the independent lifestyles of patients and requires continuous rehabilitation [6].

Due to various functional deficits and limitations in stroke survivors, rehabilitation plays an important role in improving functional recovery and minimizing disability by providing a progressive, goal-orientated process aimed at enabling stroke survivors to reach their optimal physical, cognitive, emotional, communicative, social, and functional activity level during the rest of their lives [2,7,8]. A walking impairment is often seen in stroke survivors, and gait recovery after a stroke is an important goal of post-stroke rehabilitation [9]. Therefore, it is recommended that rehabilitation for stroke patients should be aimed at improving not only the gait, balance, and daily activity performance but also the quality of life (QoL) because the decreased performance in daily life activities leads to helplessness, depression, and a lower QoL [10].

Since post-stroke rehabilitation for gait and balance impairments recovery must be provided, research using robotic devices to provide repetitive rehabilitation training is actively adopted [11,12,13]. Robotic assistance helps stroke patients recover their walking ability and mobility by performing repetitive and mobility-task training at a constant speed and intensity and reducing the physical burden of therapists because they do not need to manually place the paretic limbs or assist in trunk movements [11,12,13,14,15]. In a systemic review of the effects of repetitive gait training in stroke patients, gait training using various electromechanical devices resulted in improved independent walking compared to conventional gait training or treadmill use [11].

Setting goals is essential to stroke rehabilitation and has been recommended in various stroke guidelines [16,17]. A systematic review of the effects of goal setting in the rehabilitation of stroke patients concluded that goal setting contributes to patients’ self-efficacy and engagement in rehabilitation, but no rigorous findings could be made on the effects of goal setting in stroke rehabilitation due to the lack of a standard method of goal setting [18]. According to previous studies on the goal setting of stroke patients, the extent of patient involvement in the goal-setting process was unclear, and professionals were involved more in all aspects of goal setting process than patients [18]. Recently, a patient-focused or person-centered goal-setting approach, instead of a health professional-led goal-setting approach, has begun to be interesting due to increasing patient involvement [19]. Considering person-centered goal setting could improve the QoL by strengthening self-efficacy and rehabilitation motivation, it should be included in post-stroke rehabilitation [16,17]. Therefore, this study aimed to examine the effects of stroke rehabilitation using gait robot-assisted training and person-centered goal setting on mobility, activities of daily living, stroke self-efficacy, and health-related QoL in stroke patients with hemiplegia. We hypothesized that the group in the program rehabilitation using gait robot-assisted training and person-centered goal setting would show an improvement in their mobility, activities of daily living, stroke self-efficacy, and health-related QoL.

## 2. Materials and Methods

### 2.1. Study Design

An assessor-blinded quasi-experimental study design with a pre-posttest nonequivalent control group was used to examine the effects of stroke rehabilitation using gait robot-assisted training and person-centered goal setting for stroke patients with hemiplegia.

### 2.2. Participants

The target population was stroke patients getting post-stroke rehabilitation, and the accessible population of this study was hospitalized patients getting post-stroke rehabilitation from two hospitals. Two hospitals that specialized in post-stroke rehabilitation were selected to avoid contamination of the estimated causal effect of the intervention using gait robot-assisted training and person-centered goal setting. The characteristics of patients, bed size of each hospital, and stroke rehabilitation therapies of the two hospitals were similar. Participants who were admitted to the hospital with a gait robot-assisted training system were assigned to the experimental group, and those without gait robots were assigned to the control group. The sampling plan was self-selection and participants had selected the form of open recruitment (Figure 1).

The inclusion criteria of participants were as follows: patients aged above 40 years; patients with the first episode of hemiplegia caused by cerebral infarction or intracerebral hemorrhage; patients who had passed the acute stage of stroke; patients with a score of 1 to 3 in the functional ambulation categories (FAC); and patients with a score of 20 or higher on the Korean Version of Mini-Mental State Examination (K-MMSE) on electronic medical record system. The exclusion criteria were as follows: patients with a score of 0 (non-functional ambulator who could not walk at all) and a score of 4 to 5 (independent ambulator who could walk freely) in the FAC; patients with orthopedic problems that might interfere with walking; patients with underlying neurological disorders such as dementia; and patients with visual or auditory impairment.

Sample size estimation in the t-test, the significant level (α) = 0.05, power (1-β) = 0.080, and the medium effect size (d) = 0.50 were calculated. As a result, 27 patients were required per group. A total of 63 participants were recruited. Three participants were excluded, two participants with cognitive problems (K-MMSE < 20) and one participant who could not walk. Finally, 60 patients participated in this study.

### 2.3. Measurements

The demographic characteristics of participants were assessed: gender, age, education, job status, marital status, and subjective socioeconomic status. The disease-related characteristics of participants investigated through an electronic medical record system were assessed: subtypes of stroke, stroke-related operation history, underlying disease, medications, numeric rating scale score of pain, and K-MMSE (Korean version of Mini-Mental State Examination). The questionnaire consisted of items with Functional Ambulation Categories, a Timed Up and Go test, the Korean version of the Berg Balance Scale, the Korean version of the Modified Barthel Index, a 10-m Walk Test, Stroke Self-efficacy, and health-related quality of life.

#### 2.3.1. Functional Ambulation Categories (FAC)

The FAC are used to assess ambulation ability. This functional walking test on a six-point scale assessed ambulation status by determining how much human support the patient requires when walking: A score of 0 (non-functional ambulator who could not walk at all); A score of 1 (ambulator with physical assistance level I who requires continuous manual contact to support body weight as well as to maintain balance or to assist coordination); A score of 2 (ambulator with physical assistance level II who requires an intermittent or continuous light touch to assist balance or coordination); A score of 3 (ambulator under supervision who could ambulate on a level surface without manual contact of another person but requires standby guarding of one person either for safety or verbal cueing); A score of 4 (independent ambulator who could ambulate on a level surface but requires supervision to negotiate any of the following: stairs, inclines, or nonlevel surfaces); A score of 5 (independent ambulator who could walk everywhere including stairs) [20]. The inter-rater reliability of the test was high (0.90).

#### 2.3.2. Timed up and Go (TUG) Test

The TUG is a screening tool for basic mobility and balance. The participants were asked to stand from a chair, walk 3 m at a comfortable pace, turn around, walk back to the chair, and return to sit on the chair. The score interpreted that the total time required to perform the test was measured. A total time of fewer than 10 s and 11–20 s indicates independent ambulation and ambulation with little assistance, respectively [21]. Individuals who take longer than 30 s need physical assistance with transfers and generally cannot manage steps [21]. The inter-rater reliability of the test was high (0.95).

#### 2.3.3. Balance

The Korean version of the Berg Balance Scale (K-BBS) assesses sitting, standing, and static and dynamic balance [22]. The scale consisted of 14 functional tasks that focused on the ability to maintain a position and perform postural adjustments to complete functional movements. Each item was scored on a five-point Likert scale from 0 (an inability to complete the task entirely) to 4 points (an ability to complete the task). A global score is calculated out of 56 possible points. Scores of 0 to 20 represent balance impairment, 21 to 40 represent acceptable balance, and 41 to 56 represent good balance. A higher score indicated better balance. The reliability of the K-BBS was high (0.98).

#### 2.3.4. Activities of Daily Living

The Korean version of the Modified Barthel Index (K-MBI) by the Korean Society for Neurorehabilitation was used to assess the activities of daily living [23]. The scale consisted of ten detailed items related to self-care activities: grooming (5 points), bathing (5 points), feeding (10 points), toilet use (10 points), stairs (10 points), dressing (10 points), bowel control (10 points), bladder control (10 points), mobility on level surfaces (15 points), and transfers (15 points). The total score indicated the level of independence according to the disability rating criteria of the Ministry of Health and Welfare [24]: 91–99 points for minimal dependence, 75–90 points for mild dependence, 50–74 points for moderate dependence, 25–49 points for severe dependence, and 0–24 points for complete dependence. A higher score indicated greater independence. The reliability of the K-MBI was high (0.91).

#### 2.3.5. 10-m Walk Test (10 mWT)

The 10-m Walk Test (10 mWT) was conducted to measure the gait speed of stroke patients [25]. The participants traveled a total straight distance of 14 m. The participants started at a point 2 m before the starting point, and the time from the starting point to the endpoint crossed by the front foot was measured. The participants were allowed to conduct the test with a cane or orthosis if needed. A digital stopwatch that measured up to 0.01 s was used for assessment, and two trials were administered at the patient’s comfortable walking speed. The average value was used for the final analysis [25]. The inter-rater reliability of the test was high (0.90).

#### 2.3.6. Stroke Self-Efficacy

The stroke self-efficacy scale [26] developed by Fiona Jones (2007) for stroke patients was used in this study. The scale consisted of 13 items on an 11-point scale from 0 points for ‘not at all confident’ to 10 points for ‘very confident. The total score ranged from 0 to 130, and a higher score indicated greater self-efficacy. The reliability of the scale was high (0.95).

#### 2.3.7. Health-Related Quality of Life (QoL)

The Short Form 12 (SF-12) health survey, developed by Ware (1996) [27] and purchased through OPTUM^TM^, was used to evaluate health-related quality of life. The SF-12 covered eight health domains to assess physical and mental health. Physical health-related domains include physical functioning, physical role restriction, pain, and general health. Mental health-related domains include vitality, social function, emotional role restriction, and mental health [27]. The total score of this scale with 12 items ranging from 0 to 100 points and a higher score indicated a higher quality of life. The reliability of the SF-12 was high (0.89).

### 2.4. Interventions

#### 2.4.1. Development of Interventions

The stroke rehabilitation using gait robot-assisted training and person-centered goal setting was developed. A 6-week rehabilitation using gait robot-assisted training and person-centered goal setting for improving ambulation, balance, activities of daily living, and health-related quality of life was provided to the experimental group. The program consisted of daily gait robot-assisted training and individual education about stroke management. According to the contents of individual education, ‘Week 1 (Orientation and Journey of stroke survivors)’ consisted of educational content including the stroke recovery process, the importance of exercise in stroke, effective ways of gait training, nutrition, setting long-term and short-term goals, and secondary prevention. ‘Weeks 2 and 3 (Lower extremity strength exercise)’ consisted of leg exercises such as leg raises, bridges, squats, and superman positions to improve the lower extremity strength. ‘Weeks 4 and 5 (Balance exercise)’ aimed to improve balance through four exercises: Lean back and forth, Tilt the body left and right, Shifting weight from side to side while standing, Put your feet back and forth and turn your head. Lastly, ‘Week 6 (End of the program)’ consisted of discussions where participants discussed barriers and facilitators for goal achievement and program performance. The stroke education handbook, including an exercise log, was given to the experimental group to self-monitor their daily exercise and guide stroke recovery. In addition, immediate feedback was provided to improve self-efficacy. For setting person-centered goals to meet patients’ individual needs and values, a method of shared decision-making between patients and health professionals was used, such as encouraging patients to actively participate in goal setting, stressing patients’ ownership of their goals, and providing information.

#### 2.4.2. Experimental Group

The stroke rehabilitation using gait robot-assisted training and person-centered goal setting was administered to the experimental group from March to May 2021. The Morning Walk^®^ (CUREXO-UMK_MW01, Curexo, Seoul, Republic of Korea), a robot automation system for muscle reconstruction and joint motor function recovery, was used. The experimental group received gait robot-assisted training (once a day for 40 min) combined with conventional rehabilitation (once a day for 30 min) consisting of gait/balance-specific activities such as postural stability training and general gait training. A total of 70 min of combined rehabilitation was provided five days a week. Once the participant boarded the gait robot, information such as the number of steps and walking velocity was automatically recorded in the system. Additionally, patient and caregiver education regarding the stroke rehabilitation process and person-centered goal setting was provided every Saturday in the conference room with a researcher, and short- and long-term goals were reset for each participant after a review of the individual’s goal achievement. Once person-centered goals were set, then post-stroke rehabilitation, including gait robot-assisted training, was tailored to each patient. Daily exercise logs and appropriate incentives are also provided to support rehabilitation motivation.

#### 2.4.3. Control Group

The control group received conventional rehabilitation (twice a day, a total of 70 min, five days a week for six weeks) excluding robot-assisted training and goal setting.

### 2.5. Data Collection

After IRB approval, the pre-survey, a mobility test (FAC, K-BBS, TUG), the activities of the daily living test (K-MBI, 10 mWT), and a post-survey were conducted on participants in two groups. Two rehabilitation therapists and one nurse were hired for data collection. Two of the therapists were professional rehabilitation therapists with a license and working experience of more than five years in clinical practice. They completed the gait ability test and balance test. The survey was performed by a nurse with five years of clinical nursing experience. For data collection, each therapist from the two hospitals completed a mobility test (FAC, K-BBS, TUG) and activities of the daily living test (K-MBI, 10 mWT). The survey data collector visited the wards in two hospitals of two groups and conducted a questionnaire individually. The post-test was conducted by the same therapists and data collector as soon as the intervention was completed. The rehabilitation therapist and nurse are blinded to participants.

### 2.6. Data Analysis

The collected data were analyzed using the SPSS version 25.0 program. First, the demographic and disease-related characteristics of the patients were analyzed using frequency/percentage and mean/standard deviation. The Shapiro-Wilk test was conducted to test the normality of the study variables. A Chi-square test with Fisher’s exact test on categorical variables or an independent *t*-test on continuous variables was conducted to test homogeneity. Second, the effects of a gait robot-assisted rehabilitation on mobility test (FAC, K-BBS, TUG), activities of the daily living test (K-MBI, 10 mWT), stroke self-efficacy, and health-related QoL was analyzed using an independent *t*-test to compare statistical differences between the means of two groups and paired *t*-test to compare statistical differences between two time points before and after completing the program.

## 3. Results

### 3.1. Homogeneity Test of Demographic and Disease-Related Characteristics of Participants

There were no significant differences in the demographic and disease-related characteristics between the experimental group and the control group. The mean age of the participants was 63.08 years, and half of the patients graduated from middle school or lower. In addition, 76.7% of patients answered their subjective socioeconomic status as moderate level. Over half of the patients diagnosed their stroke as cerebral hemorrhage; the average K-MMSE and pain scores were 26.45 and NRS 2 points, respectively (Table 1).

### 3.2. Effects of Stroke Rehabilitation Using Gait Robot-Assisted Training and Person-Centered Goal Setting

There were statistically significant differences between the groups in FAC (t = 2.89, *p* = 0.005), K-BBS (t = 3.73, *p* < 0.001), TUG (t = −2.27, *p* = 0.027), K-MBI (t = 2.58, *p* = 0.012), 10 mWT (t = −2.10, *p* = 0.040), stroke self-efficacy (t = 2.23, *p* = 0.030), and health-related QoL (t = 4.90, *p* < 0.001) (Table 2).

## 4. Discussion

This research provides meaningful findings that stroke rehabilitation using gait robot-assisted training and person-centered goal setting effectively improves mobility, including gait ability and balance, activities of daily living, stroke self-efficacy, and health-related QoL in stroke patients with hemiplegia. Gait impairments negatively affect patients’ independent daily life [6], and long-term deterioration in daily activities causes adverse effects on the QoL of stroke patients [10]. Because stroke patients should recover their walking ability and mobility through continuous rehabilitation, person-centered goal setting is essential for successful rehabilitation [16,17]. As a result, the development of post-stroke rehabilitation using gait robot-assisted training and person-centered goal setting could be necessary to rehabilitate stroke patients successfully.

The key findings in this study are that gait ability-related scores, including FAC, 10 mWT, TUG, and K-MBI, of the experimental group are significantly higher than those of the control group. This is consistent with previous studies in which robot-assisted training combined with conventional rehabilitation leads to greater improvement of gait ability-related scores than conventional rehabilitation alone in stroke patients [28,29,30,31]. In a systematic review of the effects of repetitive training in stroke patients, robot-assisted training is effective for independent walking by repetitive, high-intensity gait therapy and improves lower extremity muscle strength of the paralyzed leg [11,12]. In this study, the experimental group received a combined intervention consisting of gait robot-assisted training (30 min) and conventional rehabilitation (40 min) for a total of 70 min of rehabilitation a day for five days a week, whereas the control group received conventional rehabilitation twice a day (total 70 min for five days a week). This finding suggests that the task-oriented/repetitive robot-assisted rehabilitation and combined self-exercise for six weeks effectively improve gait ability and mobility.

Another aspect, person-centered goal setting, may have influenced increased gait ability-related scores by improving patient participation and lower adherence to rehabilitation [18,19,20]. Several studies have demonstrated that person-centered goals have improved patient engagement and daily life activities, ambulation, and mobility in the rehabilitation process [18]. Our results fit with those of previous studies in that the gait ability-related scores of the experimental group were significantly higher than the control group. One plausible reason may be explained by shared goals in the rehabilitation process and customized post-stroke rehabilitation, including robot assistance. For example, the participants and research team actively engaged in the rehabilitation to set individual long-term goals and self-reflected on their performance. Such activities have motivated the participants and positively affected the program performance level, leading to improved gait ability and activities of daily living.

The findings of this study indicate that the balance score has significantly improved with a larger effect size in the experimental group than in the control group. However, a previous study reported conflicting results concerning robot-assisted rehabilitation in stroke patients [28]. Despite mixed findings on the balance function of robot-assisted training, it is obvious that the robot-assisted system totally supports body weight-bearing for patients with hemiplegia, which is beneficial for balance [31]. Body weight support through exoskeletal robots could give an advantage to patients with walking impairments by facilitating balance and gait recovery [31]. In addition, the experimental group received combined self-exercise, including lower extremity strength exercises and balance exercises, and an exercise log for self-monitoring their daily exercise was checked by the research team every Saturday. These findings give meaningful information that combined self-exercise and robot-assisted gait training might contribute to the improvement of balance in the experimental group.

The stroke self-efficacy was significantly more improved in the experimental group than in the control group. It is well-known that stroke self-efficacy is a predictor of functional independence, quality of life, and successful self-management [26]. In particular, a key area of post-stroke rehabilitation is patient-centered goals which are agreed upon between the patient and health professionals [19]. According to the self-efficacy theory, individuals should believe they can perform specific skills in a specific situation to achieve the desired goal [32]. The reason for their significant relationships is that the experimental group had set their own goals and actively participated in the program by continuously reviewing their goal achievement. Additionally, the encouragement, support, and immediate feedback from the researchers would have improved the motivation of the experimental group, leading to the successful achievement of goals and improvement of self-efficacy.

It is noteworthy that the health-related QoL was significantly improved in the experimental group compared to the control group. This agrees with previous studies regarding the improvement in the QoL by robot-assisted ambulation treatment [33]. The health-related QoL of stroke patients is related to the level of daily activities, functional movement, pain, and vitality [10]. Both physical and psychosocial factors must be comprehensively considered to improve health-related QoL [27]. Through face-to-face education in this study, person-centered goal setting was provided, and short- and long-term goals were reset for each participant after reviewing the individual’s goal achievement. Additionally, the participants learned combined exercises, including lower limb strength and balance exercises, that were performed independently. As a result, gait ability and activities of daily living in the experimental group are improved, and improved gait-related abilities allow functional movements, thereby increasing the QoL.

This study is meaningful as it is the first trial to identify the effects of stroke rehabilitation using gait robot-assisted training and person-centered goal setting for patients with hemiplegia. There are some limitations despite the significance of our study. First, since a relatively homogeneous and small number of hospitalized stroke patients with hemiplegia from two hospitals have participated in this study, the findings could not be generalized to all hemiplegic stroke patients. Second, there is a threat to internal validity due to a lack of randomization of participants into groups.

## 5. Conclusions

Our study has demonstrated that the experimental group receiving rehabilitation using gait robot-assisted training and person-centered goal setting has shown significant improvements in mobility tests (FAC, K-BBS, TUG), activities of the daily living test (K-MBI, 10 mWT), stroke self-efficacy, and health-related QoL compared to the control group. For successful rehabilitation of stroke patients, person-centered goals and stroke self-efficacy should be improved by setting realistic short- and long-term goals and a shared vision of the rehabilitation process. This further improves rehabilitation motivation, leading to improved health-related QoL and successful rehabilitation.

Based on the results of this study, there are some implications. First, future studies must evaluate the effects of stroke rehabilitation using gait robot-assisted training and person-centered goal setting from large- and small/medium-sized hospitals such as tertiary and general hospitals. Second, randomization should be adopted for scientifically rigorous trials. Third, when designing the intervention components, it should be tested whether one specific intervention or combined intervention integrating two different strategies is more effective.

## Figures and Tables

**Figure 1 healthcare-11-00588-f001:**
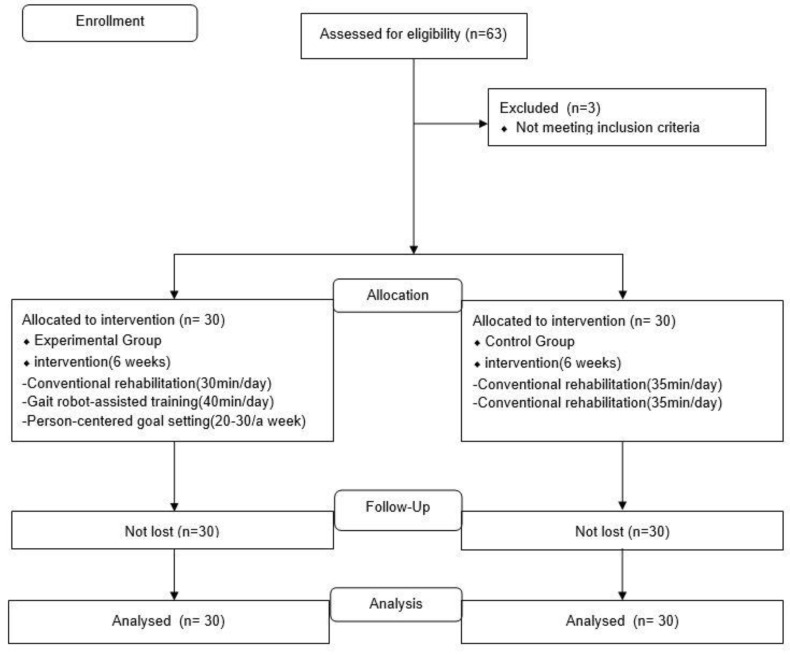
Flow Diagram of the stroke rehabilitation using gait robot-assisted training and person-centered goal setting.

**Table 1 healthcare-11-00588-t001:** Homogeneity test of demographic and disease-related characteristics of the participants.

Characteristics	Categories	Total (*n* = 60)	Exp (*n* = 30)	Cont (*n* = 30)	X^2^ or t(*p*)
Gender	Male	36 (60.0)	18 (60.0)	18 (60.0)	0.00 (1.000)
	Female	24 (40.0)	12 (40.0)	12 (40.0)	
Age(year)		63.08 ± 4.09	63.10 ± 4.34	63.07 ± 3.89	14.34 (0.350)
Education	Elementary	13 (21.7)	5 (16.6)	8 (26.6)	2.38 (0.496)
	Middle	18 (30.0)	8 (26.6)	10 (33.4)	
	High	17 (28.3)	11 (36.6)	6 (20.0)	
	College	12 (20.0)	6 (20.0)	6 (20.0)	
Job status	Yes	35 (58.3)	20 (66.6)	15 (25.0)	1.17 (0.190)
	No	25 (41.7)	10 (33.4)	15 (25.0)	
Marital status	Married	56 (93.3)	27 (90.0)	29 (96.6)	1.98 (0.370)
	Unmarried	4 (6.7)	3 (10.0)	1 (3.5)	
Subjective SES	High	8 (13.4)	3 (10.0)	5 (16.8)	2.67 (0.611)
	Moderate	46 (76.7)	25 (83.4)	21 (70.0)	
	Low	6 (10.0)	2 (6.6)	4 (13.4)	
Subtypeof stroke	Hemorrhage	27 (45)	13 (43.4)	14 (46.6)	0.07 (0.795)
	Infarction	33 (55)	17 (56.6)	16 (53.4)	
K-MMSE		26.45	26.43 ± 1.98	26.47 ± 2.37	5.90 (0.435)
Operation	Yes	8 (13.3)	4 (13.4)	4 (13.4)	0.00 (1.000)
history	No	52 (86.7)	26 (86.6)	26 (86.6)	
Underlying	Yes	31 (51.7)	16 (53.2)	15 (50.0)	0.37 (0.947)
disease	No	29 (48.3)	14 (46.6)	15 (50)	
Medication	Yes	49 (81.7)	24 (80.0)	25 (83.2)	0.34 (0.987)
	No	11 (18.3)	6 (20.0)	5 (16.6)	
NRS score		1.88 ± 0.02	1.86 ± 0.34	1.90 ± 0.31	0.21 (0.640)

Exp = experimental group; Cont = control group; SES = socioeconomic status; K-MMSE = Korean version Mini Mental State Examination; NRS = numeric rating scale of pain.

**Table 2 healthcare-11-00588-t002:** Effects of a gait robot-assisted rehabilitation using goal setting.

Variables		Groups	Pre-Test	Post-Test	Effect by Point ^†^	Intergroup Effect ^‡^
			M ± SD	M ± SD	t(*p*)	t(*p*)
Mobility	FAC	Exp	2.37 ± 0.72	3.13 ± 0.78	−7.399 (0.000) **	2.89 (0.005) **
		Cont	2.20 ± 0.55	2.73 ± 0.64	−5.113 (0.000) **	
	K-BBS	Exp	38.80 ± 5.94	46.03 ± 5.67	−13.197 (0.000) **	3.73 (<0.001) **
		Cont	37.37 ± 6.71	40.70 ± 5.39	−5.976 (0.000) **	
	TUG^4^	Exp	44.75 ± 23.90	33.41 ± 19.89	7.666 (0.000) **	−2.27 (0.027) *
		Cont	44.03 ± 25.61	40.60 ± 26.44	4.857 (0.000) **	
ADL	K-MBI	Exp	61.60 ± 15.75	73.67 ± 16.48	−8.073 (0.000) **	2.58 (0.012) *
		Cont	60.53 ± 11.72	69.00 ± 13.00	−7.429 (0.000) **	
	10 mWT	Exp	40.88 ± 23.11	31.42 ± 16.13	6.615 (0.000) **	−2.27 (0.040) *
		Cont	41.01 ± 34.95	38.83 ± 24.38	3.296 (0.003) **	
Stroke self-efficacy		Exp	59.87 ± 17.33	74.50 ± 20.23	−10.327 (0.000) **	2.23 (0.030) *
		Cont	59.57 ± 17.11	63.87 ± 10.68	−4.552 (0.000) **	
SF-12		Exp	75.02 ± 9.65	96.22 ± 10.68	−15.190 (0.000) **	4.90 (<0.001) **
		Cont	74.55 ± 10.51	83.07 ± 10.15	−6.641 (0.000) **	

* *p* < 0.05; ** *p* < 0.01; ^†^ paired *t*-test; ^‡^ independent *t*-test; Exp = experimental group; Cont = control group; FAC = Functional Ambulation Categories; K-BBS = Korean version of Berg Balance Scale; TUG = Timed Up and Go test; ADL = Activity of Daily Living; K-MBI = Korean version of Modified Barthel Index; 10 mWT = 10 meter Walking velocity Test.

## Data Availability

The datasets generated and/or analyzed during the current study are not publicly available due to concerns regarding patient privacy. The data presented in the study can be available from the corresponding author upon reasonable request.

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
