# Peer review of "Effects of Stroke Rehabilitation Using Gait Robot-Assisted Training and Person-Centered Goal Setting: A Single Blinded Pilot Study"

_healthcare, 2023, doi:10.3390/healthcare11040588_

Round 1

Reviewer 1 Report

Line 111: at section 2.3 it would be useful to include the names of the relevant tests/questionnaires that follow if you planned to use them and then continue to describe them as you do. 

Line 125: change requires to require

Line 126: change requires to require

Line 128: please rephrase "The inter-rater reliability in this study was .90" as it is not clear if it refers to [20]. Perhaps it would be better to write something like "the inter-rater relibility of the test is high (0.9) [20]" The same comment applies to lines 137, 146, 157, 166, 172, 180.

Line 184: please review sentence "The post-stroke rehabilitation using gait robot-assisted training and person-centered goal setting for stroke patients with hemiplegia developed." Check syntax

Line 185: you write"A 6-week rehabilitation using gait robot-assisted training and person-centered goal setting for improving ambulation,  balance, activities of daily living, and health-related quality of life was provided to an experimental group." an later (line 190) you include gait training only in week 1. as this is contardictory, please be clear whether gait training was included in all weeks or just one.

Line 187: change "an" to "the". The same for line 206

Line 198: you write  "In addition, immediate and constructive feedback was provided to improve self-efficacy" perhaps consider removing the phrase " and constructive" as it would be implied. Same comment for line 220.

Line 218: you write "Once person-centered goals were set, and then post-stroke rehabilitation, including robot-assisted training, was tailored to each patient." I believe "and" should be removed.

Section 2.4.3: did the experimental group also receive conventional rehabilitation twice a day? if not please comment on why. 

Line 242: you write " A Chi-square test with Fisher's exact test or t-test was conducted to test homogeneity" please explain why there was the choice of the two different tests to assess homogeneity  

Line 243: be specific on which variables you used the independent and on which the paired t test

Line 305: you write "Interestingly, our finding indicates..." Better not start with interestingly and perhaps change to "The findings of this study indicate..."

A limitations paragraph is missing from the discussion and aspects of this are included in the conclusions. 

Author Response

We sincerely thank the reviewer's valuable comments for thorough reading of our manuscript. Our manuscript has benefited from your insightful suggestions. As accepting constructive criticisms, we have carefully addressed all the comments. Our responses to reviewers’ comments are presented in red fonts and given at the file. Please see the attachment! Thanks again for your valuable comments.

Reviewer 2 Report

In their single-center study "Effects of Stroke Rehabilitation Using Gait Robot-assisted 2 Training and Person-centered Goal Setting: A Single Blinded 3 Pilot Study" Ha et al. investigate  stroke rehabilitation using gait robot-assisted training. The study is well written and the structure is clear. 

The introduction part provides sufficent information for the reader to understand the topic.

The methods part is very detailed and could be condensed to provide a clear focus for the study.

The discussion section is well organized, although I miss a "limitations section"!

Minor comments:

- Spell check for small gram. mistakes 

- Please include abbreviations

- The study could be improved by including a figure/graph

Author Response

(The authors gave the same response as above.)

Reviewer 3 Report

The author investigated the effect of rehabilitation using gait robot-assisted training and person-centered goal setting on the function recovery for patients with post-stroke hemiplegia. The study detail evaluated patient’s gait, balance, self-efficacy and quality of life. The results showed that patient received robot-assisted training and person-centered goal setting to assist rehabilitation have a better outcome than the patients received conventional rehabilitation. The study have some issue should be clarified.

Method

1. The study investigate have two interventions, robot-assisted training and person-centered goal setting, we cannot know the better outcome is related   to robot-assisted training or related to person-centered goal setting. However, the study nave completed and it cannot be changed again.

2. In introduction, author stated that setting goals is essential to stroke rehabilitation and has been recommended in various stroke guidelines. Whether the patients received conventional rehabilitation have setting goals in the study. If they have setting goals, what is the difference between the goals setting in control group and persons-centered goal setting in experimental group?

Result:

3. Table 2 should label statistical significance mark and statistical methods.

Conclusion:

4. Line 353, maybe there are some limitation not implication.

Author Response

(The authors gave the same response as above.)
